# Deep Dynamic Poisson Factorization Model

**Chengyue Gong**
Department of Information Management
Peking University
cygong@pku.edu.cn

**Win-bin Huang**
Department of Information Management
Peking University
huangwb@pku.edu.cn

## Abstract

A new model, named as deep dynamic poisson factorization model, is proposed in this paper for analyzing sequential count vectors. The model based on the Poisson Factor Analysis method captures dependence among time steps by neural networks, representing the implicit distributions. Local complicated relationship is obtained from local implicit distribution, and deep latent structure is exploited to get the long-time dependence. Variational inference on latent variables and gradient descent based on the loss functions derived from variational distribution is performed in our inference. Synthetic datasets and real-world datasets are applied to the proposed model and our results show good predicting and fitting performance with interpretable latent structure.

## 1   Introduction

There has been growing interest in analyzing sequentially observed count vectors $x_1, x_2, \ldots, x_T$. Such data appears in many real world applications, such as recommend systems, text analysis, network analysis and time series analysis. Analyzing such data should conquer the computational or statistical challenges, since they are often high-dimensional, sparse, and with complex dependence across the time steps. For example, when analyzing the dynamic word count matrix of research papers, the amount of words used is large and many words appear only few times. Although we know the trend that one topic may encourage researchers to write papers about related topics in the following year, the relationship among each time step and each topic is still hard to analyze completely.

Bayesian factor analysis model has recently reached success in modeling sequentially observed count matrix. They assume the data is Poisson distributed, and model the data under Poisson Factorize Analysis (PFA). PFA factorizes a count matrix, where $\Phi \in \mathbb{R}_+^{V \times K}$ is the loading matrix and $\Theta \in \mathbb{R}_+^{T \times K}$ is the factor score matrix. The assumption that $\theta_t \sim Gamma(\theta_{t-1}, \beta_t)$ is then included [1, 2] to smooth the transition through time. With property of the Gamma-Poisson distribution and Gamma-NB process, inference via MCMC is used in these models. Considering the lack of ability to capture the relationship between factors, a transition matrix is included in Poisson-Gamma Dynamical System (PGDS) [2]. However, these models may still have some shortcomings in exploring the long-time dependence among the time steps, as the independence assumption is made on $\theta_{t-1}$ and $\theta_{t+1}$ if $\theta_t$ is given. In text analysis problem, temporal Dirichlet process [3] is used to catch the time dependence on each topic using a given decay rate. This method may have weak points in analyzing other data with different pattern long-time dependence, such as fanatical data and disaster data [3].

Deep models, which are also called hierarchical models in Bayesian learning field, are widely used in Bayesian models to fit the deep relationship between latent variables. Examples of this include the nested Chinese Restaurant Process [4], nest hierarchical Dirichlet process [5], deep Gaussian process [6, 7] and so on. Some models based on neural network structure or recurrent structure is also used, such as the Deep Exponential Families [8], the Deep Poisson Factor Analysis based on RBM or SBN [9, 10], the Neural Autoregressive Density Estimator based on neural networks [11], Deep Poisson

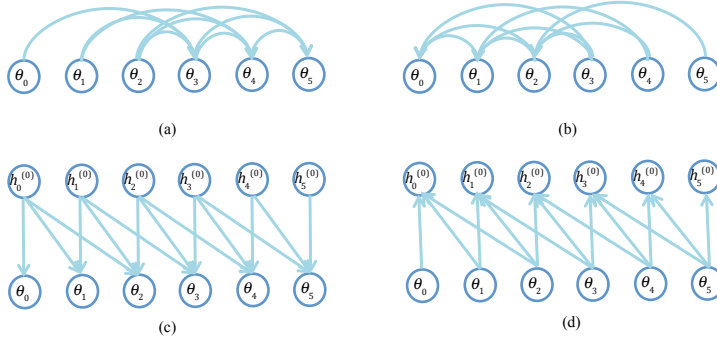

Figure 1: The visual representation of our model. In (a), the structure of one-layer model is shown. (b) shows the transmission of the posterior information. The prior and posterior distributions between interfacing layers are shown in (c) and (d).

Factor Modeling with a recurrent structure based on PFA using a Bernoulli-Poisson link [12], Deep Latent Dirichlet Allocation uses stochastic gradient MCMC [23]. These models capture the deep relationship among the shallow models, and often outperform shallow models.

In this paper, we present the Deep Dynamic Poisson Factor Analysis (DDPFA) model. Based on PFA, our model includes recurrent neural networks to represent implicit distributions, in order to learn complicated relationship between different factors among short time. Deep structure is included in order to capture the long-time dependence. An inference algorithm based on variational inference is used for inferring the latent variables. Parameters in the neural networks are learnt according to a loss function based on the variational distributions. Finally, the DDPFA model is used on several synthetic and real-world datasets, and excellent results are obtained in prediction and fitting tasks.

## 2   Deep Dynamic Poisson Factorization Model

Assume that $V$-dimensional sequentially observed count data $\boldsymbol{x}_1, \boldsymbol{x}_2, \ldots, \boldsymbol{x}_T$ are represented as a $V \times T$ count matrix $\mathbf{X}$, a count data $x_{vt} \in \{0, 1, \ldots\}$ is generated by the proposed DDPFA model as follows:

$$x_{vt} \sim Poisson(\sum_{k=1}^{K} \theta_{tk} \phi_{vk} \lambda_k \psi_v) \tag{1}$$

where the latent variables $\theta_{tk}$, $\phi_{vk}$, $\lambda_k$ and $\psi_v$ are all positive variables. $\phi_k$ represents the strength of the $k^{th}$ component and is treated as factor. $\theta_{tk}$ represents the strength of the $k^{th}$ component at the $t^{th}$ time step. Feature-wise variable $\psi_v$ captures the sparsity of the $v^{th}$ feature and $\lambda_k$ recognizes the importance of the $k^{th}$ component. According to the regular setting in [2, 13-16], the factorization is regarded as $\mathbf{X} \sim Poisson(\boldsymbol{\Phi}\boldsymbol{\Theta}^T)$. $\boldsymbol{\Lambda}$ and $\boldsymbol{\Psi}$ can be absorbed into $\boldsymbol{\Theta}$. In this paper, in order to extract the sparsity of $v^{th}$ feature or $k^{th}$ component and impose a feature-wise or temporal smoothness constraint, $\boldsymbol{\Psi}$ and $\boldsymbol{\Lambda}$ are included in our model. The additive property of the Poisson distribution is used to decompose the observed count of $x_{vt}$ as $K$ latent counts $x_{vtk}, k \in \{0, \ldots, K\}$. In this way, the model is rewritten as:

$$x_{vt} = \sum_{k=1}^{K} x_{vtk} \text{ and } x_{vtk} \sim Poisson(\theta_{tk} \phi_{vk} \lambda_k \psi_v) \tag{2}$$

Capturing the complicated temporal dependence of $\boldsymbol{\Theta}$ is the major purpose in this paper. In the previous work, transition via Gamma-Gamma-Poisson distribution structure is used, where $\boldsymbol{\theta}_t \sim Gamma(\boldsymbol{\theta}_{t-1}, \boldsymbol{\beta}_t)$ [1]. Non-homogeneous Poisson process over time to model the stochastic transition over different features is exploited in Poisson process models [17-19]. These models are then trained via MCMC or variational inference. However, it is rough for these models to catch complicated time dependence because of the weak points in their shallow structure in time dimension. In order to capture the complex time dependence over $\boldsymbol{\Theta}$, a deep and long-time dependence model with a dynamic structure over time steps is proposed. The first layer over $\boldsymbol{\Theta}$ is as follows:

$$\boldsymbol{\theta}_t \sim p(\boldsymbol{\theta}_t | \boldsymbol{h}_{t-c}^{(0)}, \ldots, \boldsymbol{h}_t^{(0)}) \tag{3}$$

where $c$ is the size of a window for analysis, and the latent variables in the $n^{th}$ layer, $n \le N$, are indicated as follows:

$$\boldsymbol{h}_t^{(n)} \sim p(\boldsymbol{h}_t^{(n)}|\boldsymbol{h}_{t-c}^{(n+1)}, ..., \boldsymbol{h}_t^{(n+1)}) \text{ and } \boldsymbol{h}_t^{(N)} \sim p(\boldsymbol{h}_t^{(N)}|\boldsymbol{h}_{t-c-1}^{(N)}, ..., \boldsymbol{h}_{t-1}^{(N)}) \quad (4)$$

where the implicit probability distribution $p(\boldsymbol{h}_t^{(n)}|\cdot)$ is modeled as a recurrent neural network. Probability AutoEncoder with an auxiliary posterior distribution $p(\boldsymbol{h}_t^{(n)}|\boldsymbol{h}_t^{(n-1)}, \ldots, \boldsymbol{h}_{t+c}^{(n-1)})$, also modeled as a neural network, is exploited in our training phase. $\boldsymbol{h}_t^{(n)}$ is a $K$-dimensional latent variable in the $n^{th}$ layer at the $t^{th}$ time step. Specially, in the $n^{th}$ layer, $\boldsymbol{h}_t^{(N)}$ is generated from a Gamma distribution with $\boldsymbol{h}_{t-c-1:t-1}^{(N)}$ as the prior information. This structure is illustrated in Figure 1.

Finally, prior parameters are placed over other latent variables for Bayesian inference. These variables are generated as: $\phi_{vk} \sim Gamma(\alpha_\phi, \beta_\phi)$ and $\lambda_k \sim Gamma(\alpha_\lambda, \beta_\lambda)$ and $\psi_v \sim Gamma(\alpha_\psi, \beta_\psi)$. Although Dirichlet distribution is often used as prior distribution [13, 14, 15] over $\phi_{vk}$ in previous works, a Gamma distribution is exploited in our model due to the including of feature-wise parameter $\psi_v$ and the purpose for obtaining feasible factor strength of $\boldsymbol{\phi}_k$.

In real world applications, like recommendation systems, the observed binary count data can be formulated by the proposed DDPFA model with a Bernoulli-Poisson link [1]. The distribution of $b$ given $\lambda$ is called Bernoulli-Poisson distribution as: $b = \mathbf{1}(x > 1), x \sim Poisson(\lambda)$ and the linking distribution is rewritten as: $f(b|x, \lambda) = e^{-\lambda(1-b)}(1 - e^{-\lambda})^b$. The conditional posterior distribution is then $(x|b, \lambda) \sim b \cdot Poisson_+(\lambda)$, where $Poisson_+(\lambda)$ is a truncated Poisson distribution, so the MCMC or VI methods can be used to do inference. Non-count real-valued matrix can also be linked to a latent count matrix via a compound Poisson distribution or a Gamma belief network [20].

## 3    Inference

There are many classical inference approaches for Bayesian probabilistic model, such as Monte Carlo methods and variational inference. In the proposed method, variational inference is exploited because the implicit distribution is regarded as prior distribution over $\boldsymbol{\Theta}$. Two stages of inference in our model are adopted: the first stage updates latent variables by the coordinate-ascent method with a fixed implicit distribution, and the parameters in neural networks are learned in the second one.

**Mean-field Approximation:** In order to obtain mean-field VI, all variables are independent and governed by its own variational distribution. The joint distribution of the variational distribution is written as:

$$q(\boldsymbol{\Theta}, \boldsymbol{\Phi}, \boldsymbol{\Psi}, \boldsymbol{\Lambda}, \boldsymbol{H}) = \prod_{v,t,n,k} q(\phi_{vk}|\phi_{vk}^*)q(\psi_k|\psi_k^*)q(\theta_{tk}|\theta_{tk}^*)q(\lambda_k|\lambda_k^*)q(h_{tk}^{(n)}|h_{tk}^{(n)*}) \quad (5)$$

where $y^*$ represents the prior variational parameter of the variable $y$. The variational parameters $\nu$ are fitted to minimize the KL divergence:

$$\nu = argmin_{\nu^*} KL(p(\boldsymbol{\Theta}, \boldsymbol{\Phi}, \boldsymbol{\Psi}, \boldsymbol{\Lambda}, \boldsymbol{H}|\boldsymbol{X})||q(\boldsymbol{\Theta}, \boldsymbol{\Phi}, \boldsymbol{\Psi}, \boldsymbol{\Lambda}, \boldsymbol{H}|\nu)) \quad (6)$$

The variational distribution $q(\cdot|\nu^*)$ is then used as a proxy for the posterior. The objective actually is equal to maximize the evidence low bound (ELBO) [19]. The optimization can be performed by a coordinate-ascent method or a variational-EM method. As a result, each variational parameter can be optimized iteratively while the remaining parameters of the model are set to fixed value. Due to Eq. 2, the conditional distribution of $(x_{vt1}, \ldots, x_{vtk})$ is a multinomial while its parameter is normalized set of rates [19] and formulated as:

$$(x_{vt1}, \ldots, x_{vtk})|\boldsymbol{\theta}_t, \boldsymbol{\phi}_v, \boldsymbol{\lambda}, \psi_v \sim Mult(x_{vt\cdot}; \boldsymbol{\theta}_t \boldsymbol{\phi}_v \boldsymbol{\lambda} \phi_v / \sum_k \theta_{tk}\phi_{vk}\lambda_k\psi_v) \quad (7)$$

Given the auxiliary variables $x_{vtk}$, the Poisson factorization model is a conditional conjugate model. The complete conditional of the latent variables is Gamma distribution and shown as:

$$\phi_{vk}|\boldsymbol{\Theta}, \boldsymbol{\Lambda}, \boldsymbol{\Psi}, \alpha, \beta, \boldsymbol{X} \sim Gamma(\alpha_\phi + x_{v\cdot k}, \beta_\phi + \lambda_k \psi_v \theta_{\cdot k})$$

$$\lambda_k|\boldsymbol{\Theta}, \boldsymbol{\Phi}, \boldsymbol{\Phi}, \alpha, \beta, \boldsymbol{X} \sim Gamma(\alpha_\lambda + x_{\cdot\cdot k}, \beta_\theta + \theta_{\cdot k} \sum_v \psi_v \phi_{vk}) \quad (8)$$

$$\psi_v|\boldsymbol{\Theta}, \boldsymbol{\Lambda}, \boldsymbol{\Phi}, \alpha, \beta, \boldsymbol{X} \sim Gamma(\alpha_\psi + x_{v\cdot\cdot}, \beta_\psi + \sum_k \lambda_k \phi_{vk} \theta_{\cdot k})$$

Generally, these distributions are derived from conjugate properties between Poisson and Gamma distribution. The posterior distribution of $\theta_{tk}$ described in Eq. 3 can be a Gamma distribution while the prior $\boldsymbol{h}_{t-c:t}^{(0)}$ is given as:

$$\theta_{tk}|\boldsymbol{\Psi}, \boldsymbol{\Lambda}, \boldsymbol{\Phi}, \boldsymbol{h}^{(0)}, \beta, \boldsymbol{X} \sim Gamma(\alpha_{\theta_{tk}} + x_{v\cdot k}, \beta_\theta + \lambda_k \sum_v \psi_v \phi_{vk}) \tag{9}$$

where $\alpha_{\theta_{tk}}$ is calculated through a recurrent neural network with $(\boldsymbol{h}_{t-c}^{(0)}, ..., \boldsymbol{h}_t^{(0)})$ as its inputs. Then the posterior distribution of $\boldsymbol{h}_{tk}^{(0)}$ described in Eq. 4 is given as:

$$h_{tk}^{(0)}|\boldsymbol{\Theta}, \boldsymbol{h}^{(1)}, \beta, \boldsymbol{X} \sim Gamma(\alpha_{h_{tk}^{(0)}} + \gamma_{h_{tk}^{(0)}}, \beta_h) \tag{10}$$

where $\alpha_{h_{tk}^{(n)}}$ is the prior information given by the $(n+1)^{th}$ layer, $\gamma_{h_{tk}^{(n)}}$ is the posterior information given by the $(n-1)^{th}$ layer. Here, the notation $h_{tk}^{(-1)}$ is equal to $\theta_{tk}$. $\alpha_{h_{tk}^{(n)}}$ is calculated through a recurrent neural network using $(\boldsymbol{h}_{t-c}^{(n+1)}, ..., \boldsymbol{h}_t^{(n+1)})$ as its inputs. $\gamma_{h_{tk}^{(n)}}$ is calculated through a recurrent neural network using $(\boldsymbol{h}_{t+c}^{(n-1)}, ..., \boldsymbol{h}_t^{(n-1)})$ as its inputs. Therefore, the distribution mentioned in Eq. 9 can be regarded as an implicit conditional distribution of $\theta_{tk}$ given $(\boldsymbol{h}_{t-c}^{(0)}, ..., \boldsymbol{h}_t^{(0)})$. And the distribution in Eq. 10 is an implicit distribution of $\alpha_{h_{tk}^{(n)}}$ given $(\boldsymbol{h}_{t-c}^{(n+1)}, ..., \boldsymbol{h}_t^{(n+1)})$ and $(\boldsymbol{h}_{t+c}^{(n-1)}, ..., \boldsymbol{h}_t^{(n-1)})$.

**Variational Inference:** Mean field variational inference can approximate the latent variables while all parameters of a neural network are given. If the observed data satisfies $x_{vt} > 0$, the auxiliary variables $x_{vtk}$ can be updated by:

$$\begin{aligned} x_{vtk} \propto exp\{&\Psi(\theta_{tk}^{shp}) - log\theta_{tk}^{rte} + \Psi(\lambda_k^{shp}) - log\lambda_k^{rte} \\ &+ \Psi(\phi_{vk}^{shp}) - log\phi_{vk}^{rte} + \Psi(\psi_v^{shp}) - log\psi_v^{rte}\} \end{aligned} \tag{11}$$

where $\Psi(\cdot)$ is the digamma function. Variables with the superscript "shp" indicate the shape parameter of Gamma distribution, and those with the superscript "rte" are the rate parameter of it. This update comes from the expectation of the logarithm of a Gamma variable as $\langle log\theta \rangle = \Psi(\theta^{shp}) - log(\theta^{rte})$. Here, $\theta$ is generated from a Gamma distribution and $\langle \cdot \rangle$ represents the expectation of the variable. Calculation of the expectation of the variable, obeyed Gamma distribution, is noted as $\langle \theta \rangle = \theta^{shp}/\theta^{rte}$. Variables can be updated by mean-field method as:

$$\begin{aligned} \phi_{vk} &\sim Gamma(\alpha_\phi + \langle x_{v\cdot k} \rangle, \beta_\phi + \langle \lambda_k \rangle \langle \psi_v \rangle \langle \theta_{\cdot k} \rangle) \\ \lambda_k &\sim Gamma(\alpha_\lambda + \langle x_{\cdot\cdot k} \rangle, \beta_\theta + \langle \theta_{\cdot k} \rangle \sum_v \langle \psi_v \rangle \langle \phi_{vk} \rangle) \\ \psi_v &\sim Gamma(\alpha_\psi + \langle x_{v\cdot\cdot} \rangle, \beta_\psi + \sum_k \langle \lambda_k \rangle \langle \phi_{vk} \rangle \langle \theta_{\cdot k} \rangle) \end{aligned} \tag{12}$$

The latent variables in the deep structure can also be updated by mean-field method:

$$\theta_{tk} \sim Gamma(\alpha_{\theta_{tk}} + \langle x_{v\cdot k} \rangle, \beta_\theta + \langle \lambda_k \rangle \sum_v \langle \psi_v \rangle \langle \phi_{vk} \rangle) \tag{13}$$

$$h_{tk}^{(n)} \sim Gamma(\alpha_{h_{tk}^{(n)}} + \gamma_{h_{tk}^{(n)}}, \beta_h) \tag{14}$$

where $\boldsymbol{\alpha}_{h_t^{(n)}} = f_{feed}(\langle \boldsymbol{h}^{n+1} \rangle), \boldsymbol{\alpha}_{h_t^{(N)}} = f_{feed}(\langle \boldsymbol{h}_{t-c-1:t-1}^N \rangle)$ and $\boldsymbol{\gamma}_{h_t^{(n)}} = f_{back}(\langle \boldsymbol{h}^{n-1} \rangle), \boldsymbol{\gamma}_{h_t^{(N)}} = f_{back}(\langle \boldsymbol{h}_{t+c+1:t+1}^N \rangle)$. $f_{feed}(\cdot)$ is the neural network constructing the prior distribution and $f_{back}(\cdot)$ is the neural network constructing the posterior distribution.

**Probability AutoEncoder:** This stage of the inference is to update the parameters of the neural networks. The bottom layer is used by us as an example. Given all latent variables, these parameters can be approximated by $p(\boldsymbol{\theta}_t|\boldsymbol{h}_{t-c}^{(0)}, ..., \boldsymbol{h}_t^{(0)})$ and $p(\boldsymbol{h}_t^{(0)}|\boldsymbol{\theta}_{t+c}, ..., \boldsymbol{\theta}_t)$. $p(\boldsymbol{\theta}_t^{(n)}|\boldsymbol{h}_{t-c}^{(0)}, ..., \boldsymbol{h}_t^{(0)}) = Gamma(\boldsymbol{\alpha}_{\theta_t}, \boldsymbol{\beta}_h)$ is modeled by a RNN with the inputs $(\boldsymbol{h}_{t-c}^{(0)}, ..., \boldsymbol{h}_t^{(0)})$ and the outputs, $\boldsymbol{\alpha}_{\theta_t}$. The

$p(\boldsymbol{h}_t^{(0)}|\boldsymbol{\theta}_{t+c}, ..., \boldsymbol{\theta}_t)$ is also modeled as a RNN with the inputs $(\boldsymbol{\theta}_{t+c}, ..., \boldsymbol{\theta}_t)$ and the outputs $\boldsymbol{\gamma}_{h_t^{(0)}}$ . With the posterior distribution from $\boldsymbol{\Theta}$ to $\boldsymbol{H}^{(0)}$ and the prior distribution from $\boldsymbol{H}^{(0)}$ to $\boldsymbol{\Theta}$, the probability of $\boldsymbol{\Theta}$ should be maximized. The loss function of these two neural networks is as follows:

$$\max_{\boldsymbol{W}}\{\int p(\boldsymbol{\Theta}|\boldsymbol{H}^{(0)})p(\boldsymbol{H}^{(0)}|\boldsymbol{\Theta})d\boldsymbol{H}^{(0)}\} \tag{15}$$

where $\boldsymbol{W}$ represents the parameters in neural networks. Because the integration in Eq. 15 is intractable, a new loss function should include auxiliary variational variables $\boldsymbol{H}^{(0)\prime}$. Assume that $\boldsymbol{H}^{(0)\prime}$ is generated by $\boldsymbol{\Theta}$, the optimization can be regarded as maximizing the probability of $\boldsymbol{\Theta}$ with minimal difference between $\boldsymbol{H}^{(0)\prime}$ and $\boldsymbol{H}^{(0)}$ as $\max_{\boldsymbol{W}}\{p(\boldsymbol{\Theta}|\boldsymbol{H}^{(0)})\}$ and $\min_{\boldsymbol{W}}\{KL(p(\boldsymbol{H}^{(0)\prime}|\boldsymbol{\Theta})||p(\boldsymbol{H}^{(0)}|\boldsymbol{H}^{(1)}))\}$

Then approximating the variables generated from a distribution by its expectation, the loss function, similar to variational AutoEncoder [21], can be simplified to:

$$\min_{\boldsymbol{W}}\{\|\langle p(\boldsymbol{H}^{(0)\prime}|\boldsymbol{\Theta})\rangle - \langle p(\boldsymbol{H}^{(0)}|\boldsymbol{H}^{(1)})\rangle\|_2 + \|\boldsymbol{\Theta} - \langle p(\boldsymbol{\Theta}|\boldsymbol{H}^{(0)})\rangle\|_2\} \tag{16}$$

Since only a few samples are drawn from one certain distrbution, which means sampling all latent variables is high-cost and useless, differentiable variational Bayes is not suitable. As a result, we focus more on fitting data than generating data. In our objective, the first term, a regularization, encourages the data to be reconstructed from the latent variables, and the second term encourages the decoder to fit the data.

The parameters in the networks for $n^{th}$ and $(n+1)^{th}$ layer are trained by the loss function:

$$\min_{\boldsymbol{W}}\{\|\langle p(\boldsymbol{H}^{(n+1)\prime}|\boldsymbol{H}^{(n)})\rangle - \langle p(\boldsymbol{H}^{(n)}|\boldsymbol{H}^{(n+1)})\rangle\|_2 \\ + \|\boldsymbol{H}^{(n)} - \langle p(\boldsymbol{H}^{(n)}|\boldsymbol{H}^{(n+1)})\rangle\|_2\} \tag{17}$$

In order to make the convergence more stable, the term of $\boldsymbol{\Theta}$ in the first layer is collapsed into $\boldsymbol{X}$ by using the fixed latent variables approximated by mean-field VI, and the loss function is as follows:

$$\min_{\boldsymbol{W}}\{\|\langle p(\boldsymbol{H}^{(0)\prime}|\boldsymbol{\Theta})\rangle - \langle p(\boldsymbol{H}^{(0)}|\boldsymbol{H}^{(1)})\rangle\|_2 + \|\boldsymbol{X} - \langle\boldsymbol{\Psi}\rangle\langle\boldsymbol{\Lambda}\rangle\langle\boldsymbol{\Phi}\rangle\langle p(\boldsymbol{\Theta}|\boldsymbol{H}^{(0)})\rangle\|_2\} \tag{18}$$

After the layer-wise training, all the parameters in neural networks are jointly trained by the fine-tuning trick in stacked AutoEncoder [22].

## 4 Experiments

In this section, four multi-dimensional synthetic datasets and five real-world datasets are exploited to examine the performance of the proposed model. Besides, the results of three existed methods, PGDS, LSTM, and PFA, are compared with results of our model. PGDS is a dynamic Poisson-Gamma system mentioned in Section 1, and LSTM is a classical time sequence model. In order to prove the deep relationship learnt by the deep structure can improve the performance, a simple PFA model is also included as a baseline.

All hyperparameters of PGDS set in [2] are used in this paper. 1000 times gibbs sampling iterations for PGDS is performed, 100 iterations used mean-field VI for PFA is performed, and 400 epochs is executed for LSTM. The parameters in the proposed DDPFA model are set as follows:$\alpha_{(\lambda,\phi,\psi)} = 1, \beta_{(\lambda,\phi,\psi)} = 2, \alpha_{(\theta,h)} = 1, \beta_{(\theta,h)} = 1$. The iterations is set to 100. The stochastic gradient descent for the neural networks is executed 10 epochs in each iteration. The size of the window is 4. Hyperparameters of PFA are set as the same to our model. Data in the last time step is exploited as the predicting target in a prediction task. Mean squared error (MSE) between the ground truth and the estimated value and the predicted mean squared error (PMSE) between the ground truth and the predicted value in next time step are exploited to evaluate the performance of each model.

### 4.1 Synthetic Datasets

The multi-dimensional synthetic datasets are obtained by using the following functions where the subscript stands for the index of dimension:

Table 1: The result on the synthetic data

| Data | Measure | DDPFA | PGDS | LSTM | PFA |
|------|---------|-------|------|------|-----|
| SDS1 | MSE | $\mathbf{0.15} \pm 0.01$ | $1.48 \pm 0.00$ | $2.02 \pm 0.23$ | $1.61 \pm 0.00$ |
|      | PMSE | $\mathbf{2.07} \pm 0.02$ | $5.96 \pm 0.00$ | $2.94 \pm 0.31$ | - |
| SDS2 | MSE | $\mathbf{0.06} \pm 0.01$ | $3.38 \pm 0.00$ | $1.83 \pm 0.04$ | $4.42 \pm 0.00$ |
|      | PMSE | $\mathbf{2.01} \pm 0.02$ | $3.50 \pm 0.01$ | $2.41 \pm 0.06$ | - |
| SDS3 | MSE | $\mathbf{0.10} \pm 0.02$ | $1.62 \pm 0.00$ | $1.13 \pm 0.06$ | $1.34 \pm 0.00$ |
|      | PMSE | $\mathbf{2.14} \pm 0.04$ | $4.33 \pm 0.01$ | $3.03 \pm 0.05$ | - |
| SDS4 | MSE | $\mathbf{0.15} \pm 0.03$ | $2.92 \pm 0.00$ | $4.30 \pm 0.26$ | $0.25 \pm 0.00$ |
|      | PMSE | $\mathbf{1.48} \pm 0.04$ | $6.41 \pm 0.01$ | $4.67 \pm 0.24$ | - |

SDS1:$f_1(t) = f_2(t) = t, f_3(t) = f_4(t) = t + 1$ on the interval $t = [1 : 1 : 6]$.

SDS2:$f_1(t) = t \quad (mod \quad 2), f_2(t) = 2t \quad (mod \quad 2) + 2, f_3(t) = t$ on the interval $t = [1 : 1 : 20]$.

SDS3:$f_1(t) = f_2(t) = t, f_3(t) = f_4(t) = t + 1, f_5(t) = \boldsymbol{I}(4|t)$ on the interval $t = [1 : 1 : 20]$, where $\boldsymbol{I}$ is an indicator function.

SDS4:$f_1(t) = t \quad (mod \quad 2), f_2(t) = 2t \quad (mod \quad 2) + 2, f_3(t) = t \quad (mod \quad 10)$ on the interval $t = [1 : 1 : 100]$.

The number of factor is set to $K = 3$, and the number of the layers is 2. Both fitting and predicting tasks are performed in each model. The hidden layer of LSTM is 4 and the size in each layer is 20. In Table 1, it is obviously that DDPFA has the best performance in fitting and prediction task of all the datasets. Note that the complex relationship learnt from the time steps helps the model catch more time patterns according to the results of DDPFA, PGDS and PFA. LSTM performs worse in SDS4 because the noise in the synthetic data and the long time steps make the neural network difficult to memorize enough information.

## 4.2 Real-world Datasets

Five real-world datasets are used as follows:

*Integrated Crisis Early Warning System (ICEWS)*: ICEWS is an international relations event data set extracted from news corpora used in [2]. We therefore treated undirected pairs of countries $i \leftrightarrow j$ as features and created a count matrix for the year 2003. The number of events for each pair during each day time step is counted, and all pairs with fewer than twenty-five total events is discarded, leaving $T = 365, V = 6197$, and 475646 events for the matrix.

*NIPS corpus (NIPS)*: NIPS corpus contains the text of every NIPS conference paper from the year 1987 to 2003. We created a single count matrix with one column per year. The dataset is downloaded from Gal's page [1], with $T = 17, V = 14036$, with 3280697 events for the matrix.

*Ebola corpus (EBOLA)*[2] : EBOLA corpus contains the data for the 2014 Ebola outbreak in West Africa every day from Mar 22th, 2014 to Jan 5th 2015, each column represents the cases or deaths in a West Africa country. After data cleaning, the dataset is with $T = 122, V = 16$.

*International Disaster(ID)*[3] : The International Disaster dataset contains essential core data on the occurrence and effects of over 22,000 mass disasters in the world from 1900 to the present day. A count matrix with $T = 115$ and $V = 12$ is built from the events of disasters occurred in Europe from the year 1902 to 2016, classified according to their disaster types.

*Annual Sheep Population(ASP)*[4] : The Annual Sheep Population contains the sheep population in England & Wales from the year 1867 to 1939 yearly. The data matrix is with $T = 73, V = 1$.

Table 2: The result on the real-world data

| Data | Measure | DDPFA | PGDS | LSTM | PFA |
|---|---|---|---|---|---|
| ICEWS | MSE | **3.05** $\pm$ 0.02 | 3.21 $\pm$ 0.01 | 4.53 $\pm$ 0.04 | 3.70 $\pm$ 0.01 |
|  | PMSE | **0.96** $\pm$ 0.03 | **0.97** $\pm$ 0.02 | 6.30 $\pm$ 0.03 | - |
| NIPS | MSE | **51.14** $\pm$ 0.03 | 54.71 $\pm$ 0.08 | 1053.12 $\pm$ 39.01 | 69.05 $\pm$ 0.43 |
|  | PMSE | **289.21** $\pm$ 0.02 | 337.60 $\pm$ 0.10 | 1728.04 $\pm$ 38.42 | - |
| EBOLA | MSE | **381.82** $\pm$ 0.13 | 516.57 $\pm$ 0.01 | 4892.34 $\pm$ 10.21 | 1493.32 $\pm$ 0.21 |
|  | PMSE | **490.32** $\pm$ 0.12 | 1071.01 $\pm$ 0.01 | 5839.26 $\pm$ 11.92 | - |
| ID | MSE | **1.59** $\pm$ 0.01 | 3.45 $\pm$ 0.00 | 11.19 $\pm$ 1.32 | 4.41 $\pm$ 0.01 |
|  | PMSE | **5.18** $\pm$ 0.01 | 10.44 $\pm$ 0.00 | 10.37 $\pm$ 1.54 | - |
| ASP | MSE | **14.17** $\pm$ 0.02 | 2128.47 $\pm$ 0.02 | 17962.47 $\pm$ 14.12 | 388.02 $\pm$ 0.01 |
|  | PMSE | **21.23** $\pm$ 0.04 | 760.42 $\pm$ 0.02 | 21324.72 $\pm$ 17.48 | - |

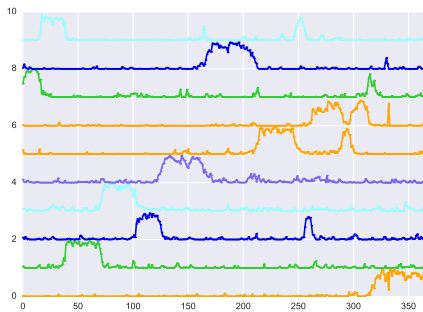
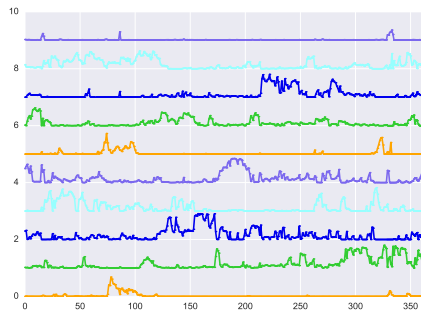

(a) PGDS  (b) DDPFA

Figure 2: The visual of the factor strength in each time step of the ICEWS data, the data is normalized each time step. In (a), the result of PGDS shows the factors are shrunk to some local time steps. In (b), the result of DDPFA shows the factors are not taking effects locally.

We set $K = 3$ for ID and ASP datasets, while set $K = 10$ for the others. The size of the hidden layers of the LSTM is 40. The settings of remainder parameters here are the same as those in the above experiment. The results of the experiment are shown in Table 2.

Table 2 shows the results of four different models on the five datasets, and the proposed model DDPFA has satisfying performance in most experiments although the DDPFA's result in ICEWS prediction task is not good enough. While smoothed data obtained from the transition matrix in PGDS performs well in this prediction task. However, In EBOLA and ASP datasets, PGDS fails in catching complicated time dependence. And it is a tough challenge for LSTM network to memorize enough useful patterns while its input data includes long-time patterns or the dimension of the data is particular high.

According to the observation in Figure 2, it can be shown that the factors learnt by our model are not activated locally compared to PGDS. Natrually, in real-world data, it is impossible that only one factor happens in one time step. For example, in the ICEWS dataset, the connection between Israel and Occupied Palestinian Territory still remains strong during the Iraq War or other accidents. Figure 2(a) reveals that several factors at a certain time step are not captured by PGDS. In Figure 3, the changes of two meaningful factors in ICEWS is shown. These two factor, respectively, indicate Israel-Palestinian conflict and six-party talks. The long-time activation of factors is shown in thi figure, since DDPFA model can capture weak strength along time.

In Table 3, we show the performance of our model with different sizes. From the table, performance cannot be improved distinctly by adding more layers or adding more variables in upper layer. It is also noticed that expanding the dimension in bottom layer is more useful than in upper layers. The results reveal two problems of proposed DDPFA: "pruning" and uselessness of adding network layers.

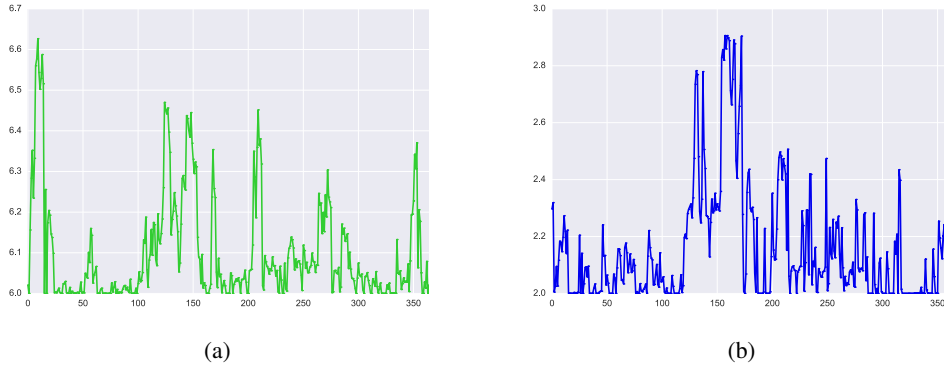

<div align="center">(a)                (b)</div>

Figure 3: The visual of the top two factors of the ICEWS data generated by DDPFA method. In (a), 'Japan–Russian Federation', 'North Korea–United States', 'Russian Federation–United States', 'South Korea–United States', and 'China–Russian Federation' are the largest features due to their loading weights. This factor stands for six-party talks and other accidents about it. In (b), 'Israel–Occupied Palestinian Territory', 'Israel–United States', 'Occupied Palestinian Territory–United States' are the largest features and it stands for the Israeli-Palestinian conflict.

Table 3: MSE on real datasets with different sizes.

| Size | ICEWS | NIPS | EBOLA |
|---|---|---|---|
| 10-10-10 | 2.94 | 51.24 | 382.17 |
| 10-10-10 (ladder structure) | 2.88 | 49.81 | 379.08 |
| 10-10 | 3.05 | 51.14 | 381.82 |
| 32-32-32 | 2.95 | 50.12 | 379.64 |
| 32-32-32 (ladder structure) | 2.86 | 49.26 | 377.81 |
| 32-64-64 | 2.93 | 50.18 | 380.01 |
| 64-32-32 | 2.90 | 50.04 | 378.87 |

[25] notices hierarchical latent variable models do not take advantage of the structure, and gives such a conclusion that only using the bottom latent layer of hierarchical variational autoencoders should be enough. In order to solve this problem, the ladder-like architecture, in which each layer combines independent variables with latent variables depend on the upper layers, is used in our model. It is noticed that using ladder architecture could reach much better results from Table 3. Another problem, "pruning", is a phenomenon where the optimizer severs connections between most of the latent variables and the data [24]. In our experiments, it is noticed that some dimenisions in the latent layers only contain data noise. This problem is also found in differentiable variational Bayes and solved by using auxiliary MCMC strcuture [24]. Therefore, we believe this problem is caused by MF-variational inference used in our model and we hope it can be solved if we try other inference methods.

## 5 Summary

A new model, called DDPFA, is proposed to obtain long-time and complicated dependence in time series count data. Inference in DDPFA is based on variational method for estimating the latent variables and approximating parameters in neural networks. In order to show the performance of the proposed model, four multi-dimensional synthetic datasets and five real-world datasets, ICEWS, NIPS corpus, EBOLA, International Disaster and Annual Sheep Population, are used, and the performance of three existed methods, PGDS, LSTM, and PFA, are compared. According to our experimental results, DDPFA has better effectivity and interpretability in sequential count analysis.

## Footnotes

[1] http://ai.stanford.edu/gal/data.html

[2] https://github.com/cmrivers/ebola/blob/master/country_timeseries.csv

[3] http://www.emdat.be/

[4] https://datamarket.com/data/list/?q=provider:tsdl

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
