[Reviews · NeurIPS 2017]

Reviewer 1



This papers introduces the deep dynamic Poisson factorization model, a model that builds on PF to allow for temporal dependencies. In contrast to previous works on dynamic PF, this paper uses a simplified version of a recurrent neural network to allow for long-term dependencies. Inference is carried out via variational inference, with an extra step to maximize over the neural network parameters. The paper reports experiments on 5 real-world datasets. Overall, I found the idea potentially interesting, but I still think the paper needs some improvements in its execution before it can be accepted in a conference like NIPS. Please find my comments below. +The formulation in Eq. 4 doesn't correspond to a RNN structure, because h_t^{(n)} depends only on the top-layer memory vectors, and it does not depend on previous time steps for the same layer, as in the RNN setup. Is there any reason for this simplified structure? +In Section 2, some of the model equations are missing (e.g., it is not clear how to parameterize the Gamma in Eq. 4, or what's the distribution that generates theta_t in Eq. 3). +I disagree with the discussion in lines 110-118 regarding "implicit distributions". Eqs. 9 and 10 are actually explicit: they are Gammas with given shapes and rates. Since the distribution of theta_tk (Eq. 9) can be computed in closed form conditioned on h^{(0)}, it cannot be called implicit. The same comment applies to h in Eq. 10. +Lines 30-31: The sentence "This method may have weak points in analyzing other data with different pattern long-time dependence, such as fanatical data and disaster data" wasn't clear to me; can you elaborate? +The subsection in lines 133-155 isn't clear to me. What is exactly the "loss function" in Eq. 15? +MSE/PMSE is typically not a good metric for PF-like models. The results should additionally include at least predictive log-likelihood. +The value of K in the experiments (line 207) seems too small to me. I suggest the authors use larger values, such as K=100 (at least). +I couldn't connect the discussion in lines 224-232 with Fig. 3: it is not clear. +The update equations for the variational inference procedure (Eqs. 11-14) can safely be moved to the supplement. +The caption in Fig. 4 cannot be read clearly. +I *strongly* recommend to follow general writing advice about the use of the passive voice (specially for the abstract). +The writing quality is poor, and the paper contains many typos. Here are some examples: -Line 15, "high-dimensional" -Line 22, "factorize" -Line 24, "property of the" -Lines 29, 34, "Dirichlet" -Line 34, "nested" -Fig. 1, "visual representation" -Fig.1, "transmission" (not sure if that's an appropriate word anyways) -Incomplete sentences (e.g., line 49) -Many missing articles (e.g., "Although Dirichlet distribution is often used as prior distribution")

Reviewer 2



The authors propose a novel method to capture the complex long-time dependence over the count vectors by exploiting recurrent neural networks to represent implicit distributions, which is realized by the gamma distribution. The shape parameters of those gamma distributions, capturing the time dependence, are approximated by neural networks, which try to cover the information from prior (from higher layer) and likelihood (from low layer). The parameters in neural network are updated based on the expectation of latent variables, which is helpful to reduce the variance caused by sampling and increase the computational efficiency. The model provides an interesting and smart way to handle the long dependency problem in the dynamic model. I like it, but this manuscript may be written a little rush and I I have some comments as below: 1. Some critical typos, i.e. the equation 10. 2. The authors consider the scale parameter as fixed and only update the shape parameter in those gamma distributions, which need some detailed discussion and analysis. 3. In the experiments, the authors only show the performance of the proposed model with two layers and do not compare it with the model with the single layer and more layers, which is not enough to understand the influence of the layers on the performance. Some other parameters also need discussing, i.e. the size of the window, the number of factors. 4. For better understanding, I suggest the authors display the detailed graphical illustrations of the whole model. According to the problem, the author can refer to the figure 1 in [1]. 5. Recently some new deep Poisson factor analysis models have been proposed, such as [2] and [3], which need to discuss in Introduction. [1] Chung J, Kastner K, Dinh L, et al. A Recurrent Latent Variable Model for Sequential Data. NIPS, 2015. [2] Mingyuan Zhou, Yulai Cong, and Bo Chen, Augmentable gamma belief networks, Journal of Machine Learning Research,17(163), 1-44, 2016. [3] Yulai Cong, Bo Chen, Hongwei Liu, and Mingyuan Zhou, Deep latent Dirichlet allocation with topic-layer-adaptive stochastic gradient Riemannian MCMC, ICML 2017.

Reviewer 3



The paper deals with incorporating long range temporal dependence in a Poisson factorization model with deep structure. As opposed to the previous works on dynamic PF, this paper employs a recurrent neural network like structure for modeling the long-term dependency. Finally, the inference is carried out using variational inference with some optimization routines for finding the parameters of the neural network. The paper has good technical contents but is also plagued with some grammatical and factual errors. These are listed below: 1. The statement made about modeling nonnegative numbers on lines 86-87 is incorrect. It's the Poisson Randomized Gamma distribution that is used for modeling the nonnegative numbers. 2. Please make the best results bold in Table 2. 3. Held-out perplexity should have been a stronger indicator in the experiments in Section 4.2, though MSE/PMSE are okay. 4. The limitations in the existing works for modeling the long-range dependency (reference 1,2) are adequately addressed in this paper. However, to illustrate the utility of the proposed method with multiple levels of hierarchy, a comparison with dynamic PF with one level deep architecture could have been useful. The comparison with LSTM is useful, but the learning algorithms are different. Therefore, this comparison does not truly justify the need for multiple levels of hierarchy in the proposed method. 5. There are several grammatical errors and typos as listed below that make the paper little difficult to read: -- "unit the data" -- "Noticing the lack of capturing .." -- "PFA factorized a count matrix" (the tense of this sentence is incongruent with the tense in other sentences in the same paragraph) -- "other data with different pattern long-time dependence .." -- "Examples for this including .." -- "shows excellent results on predict and .." -- "However, it is rough for them .." -- "V-dimesnion sequentially .." -- "K-dimension latent variable .." -- "due to the including of feature-wise" -- "recommend systems" etc.